# Additional Tissue Sampling Trials Did Not Change Our Thyroid Practice

**DOI:** 10.3390/cancers13061270

**Published:** 2021-03-12

**Authors:** Hisakazu Shindo, Kennichi Kakudo, Keiko Inomata, Yusuke Mori, Hiroshi Takahashi, Shinya Satoh, Hiroyuki Yamashita

**Affiliations:** 1Department of Surgery, Yamashita Thyroid Hospital, Fukuoka 812-0034, Japan; mori@kojosen.com (Y.M.); htakahashi@kojosen.com (H.T.); shinya.s.48128@kojosen.com (S.S.); yamaftc@kojosen.com (H.Y.); 2Thyroid Disease Center, Department of Pathology, City General Hospital, Osaka 594-0073, Japan; kakudo@thyroid.jp; 3Department of Pathology, Yamashita Thyroid Hospital, Fukuoka 812-0034, Japan; kinomata@kojosen.com

**Keywords:** follicular thyroid carcinoma, sampling method, capsular invasion, vascular invasion, overdiagnosis

## Abstract

**Simple Summary:**

Some studies have suggested that the use of additional tissue blocks to diagnose follicular thyroid carcinoma (FTC) could increase the accuracy of diagnosis and improve prognosis, and entire capsule sampling was recommended for a definitive diagnosis of borderline thyroid tumors in 2016. We conducted a study in 2016 to examine whether additional tissue sampling of the encapsulated follicular tumor increases the diagnosis of malignant cases in our patient cohort. Furthermore, the diagnosis was reclassified according to the 4th edition of the World Health Organization’s classification system. The additional tissue sampling only had a slight impact on our thyroid practice and resulted in no benefits to the patient; therefore, we decided to cease it.

**Abstract:**

This study aimed to determine whether additional tissue sampling of encapsulated thyroid nodules would increase the frequency of follicular thyroid carcinoma (FTC) diagnoses. We examined thyroid tissue specimens from 86 patients suspected of FTC (84.9% female; mean age, 49.0 ± 17.8 years). The number of tissue blocks created for pathological assessments ranged from 3 to 20 (mean, 9.1 ± 4.1); the numbers in the previous method recommended by the Japanese General Rules for the Description of Thyroid Cancer and additional blocks ranged from 1 to 12 (mean, 6.0 ± 2.8) and from 1 to 8 (mean, 3.1 ± 2.0), respectively. The additional blocks were subsequently examined to determine whether any diagnoses changed from those based on the previous method. Five patients were diagnosed with FTC using the previous method; however, additional tissue blocks led to the diagnosis of FTC in 6 patients, as 1 diagnosis was revised from follicular adenoma to FTC. It has been reported that increasing the number of tissue blocks used for pathological assessments can increase the frequency of FTC diagnoses; however, this was not clinically significant in thyroid carcinoma, which requires completion thyroidectomy and radioactive iodine treatment. It resulted in no benefits to the patient because all minimally invasive FTCs, follicular tumors of uncertain malignant potential (FT-UMP), and follicular adenomas are treated with lobectomy alone in Japan. Additional tissue sampling only had a slight impact on our thyroid practice; therefore, we decided to cease it.

## 1. Introduction

Follicular thyroid carcinoma (FTC) is the second most common thyroid malignancy after papillary thyroid carcinoma (PTC). In Japan, cases of FTC account for about 5% of total malignant thyroid tumors [1]. Morphologically, FTC is typically characterized as an oval-shaped, encapsulated, solid nodule on gross examination. In 2017, the classifications of FTC were revised in the 4th edition of the World Health Organization’s (WHO) Classification of Tumors of Endocrine Organs into the following three types: (1) minimally invasive FTC, in which only capsular invasion is observed; (2) encapsulated angioinvasive FTC, in which vascular invasion is observed; and (3) widely invasive FTC, in which widespread infiltration of thyroid tissue surrounding the tumor is observed. Furthermore, a significant number of minimally invasive FTCs were downgraded from malignant tumors to the borderline tumor category in instances where invasion is questionable [2,3]. Following the issuance of the 4th edition of the WHO’s classification, the 8th edition of the Japanese General Rules for the Description of Thyroid Cancer was published in 2019 to adopt the revisions made by the WHO. In this edition, FTC is also classified into three types [4], and the minimally invasive type or the encapsulated angioinvasive type of FTCs was predominant in our patient cohort. Grossing of the organ is key to an accurate pathological diagnosis. In other words, it is important to obtain enough tissue specimens containing any suspicious foci based on gross examination. The Japanese General Rules for the Description of Thyroid Cancer recommend that, in principle, formalin-fixed thyroid specimens should be sectioned sagittally, with a thickness of 3–5 mm. Although sampling of multiple blocks is considered good practice, sampling of the entire specimen is not recommended [4]. On the other hand, all Western clinical guidelines recommend entire capsule sampling of encapsulated thyroid tumors. Their main intention is to prevent cancer from being overlooked. This study was designed to validate the studies by Lang et al. and Oh et al., which reported that increasing the number of samples increased the detection rate of capsular invasion [5,6]. The purpose of this study was to discover an ideal sampling method to efficiently identify clinically significant FTCs in community hospitals for the best and adequate treatment for patients. A cost-effective clinical practice is essential and indispensable for community hospitals not only in Western countries but also in developing countries in Asia. We believe that a new, affordable approach is necessary. one which can more efficiently and accurately identify invasion in thyroid nodules and replace the entire sampling method currently recommended by the WHO classification and most Western clinical guidelines [3,7,8].

## 2. Materials and Methods

### 2.1. Patients

Prior to this study, thyroid surgery was performed for 839 cases during the period between January 2015 and December 2015. Of those, 501 surgeries were performed for malignant thyroid tumors, 22 of which were FTCs (4.4% of all cases of malignant thyroid tumors), while 235 surgeries were for benign thyroid adenomas (Table 1). The sampling method in the Yamashita Thyroid Hospital was adapted from a method recommended by the 6th edition of the Japanese General Rules for the Description of Thyroid Cancer [9]. The mean number of tissue blocks used to diagnose FTC was 5 (range, 2–9); and the mean number of blocks for benign and borderline tumor diagnoses made via unilateral lobectomy was also 5 (range, 2–9).

### 2.2. Tissue Preparation and Pathological Evaluations

This study was conducted to validate whether increasing the number of blocks used for pathological diagnoses would also lead to an increase in the frequency of FTC diagnoses. The inclusion criteria were as follows: (1) encapsulation or good demarcation of thyroid nodules larger than 1.5 cm, and (2) solid or predominantly solid nodules. Exclusion criteria included a grossly invasive thyroid nodule or cytologically proven PTC, along with predominantly cystic nodules. From March 2016 to March 2017, a total of 86 cases from 712 surgically treated thyroid nodules were included in this study. The maximum size of the tumor was taken as the size measured by ultrasonography.

All thyroid specimens were fixed in 10% buffered formalin for more than 24 h, and sections were cut sagittally into 5 to 10 slices of 3–5 mm thickness (Figure 1). Usually, one or two representative slices were selected and submitted for histopathologic examination after cutting into 2–8 blocks depending on the tumor size (the method recommended by the Japanese General Rules for the Description of Thyroid Cancer—JGRDTC— which has been implemented at our hospital). When suspicious findings were noted, such as findings suggesting invasion, additional sections were often taken. We conducted the current study between March 2016 and March 2017, and 1 or 2 extra slices were submitted for additional histological examination (as additional blocks) to examine whether this additional sampling changed our thyroid practice.

For the case shown in Figure 1, thyroid tissue was cut into eight tissue slices (Figure 1, middle), with a nodule being visible in seven of these sections. We prepared four blocks according to the JGRDTC method, comprising sections resected from the maximum cut surface (Figure 1(4a,4b)) and resected sections were selected based on gross examination (Figure 1(5a,5b)). Four additional blocks were prepared (Figure 1(2a,2b,3a,3b)) using two resected sections selected via gross examination. A tentative pathological diagnosis was made in each case, based on the JGRDTC method (the “tentative diagnosis”), and a “revised diagnosis” based on all blocks. A pathologist’s assistant, K.I., prepared the tissue blocks, and a pathologist specializing in endocrine pathology, K.K., made the pathological diagnosis. The pathological diagnosis made at the time of surgery was based on the 3rd edition of the WHO classification [2]. In this study, both the tentative and revised diagnoses were reclassified according to the 4th edition of the WHO Classification of Tumours of Endocrine Organs [3].

### 2.3. Statistical Analysis

JMP for Windows version 11.0.0 software (SAS Institute, Cary, NC, USA) was used for statistical analysis. One-way analysis of variance was performed for group comparisons. The level of significance was defined as *p* < 0.05.

## 3. Results

### 3.1. Patient Characteristics

The study included 86 patients, with a mean age of 49.0 ± 17.8 years (range, 11–81 years), with 73 patients being women (84.9%) and 13 men (15.1%). The mean size of the thyroid nodule was 4.1 ± 1.3 cm (range, 1.6–8.9 cm). Tumors with a size of less than 2.0 cm were observed in two patients (2.3%), 2.0–4.0 cm in 46 patients (53.5%), and larger than 4.0 cm in 38 patients (44.2%). In terms of surgical techniques used, unilateral thyroid lobectomy with or without isthmectomy was performed in 72 patients (83.7%) and total thyroidectomy was performed in 14 patients (16.3%). Among the 14 patients who underwent total thyroidectomy, 5 cases were found to be complicated by the presence of papillary carcinoma on the contralateral lobe (all T1N0), 6 were complicated by the presence of follicular adenoma or adenomatous goiter on the contralateral lobe, and three were complicated by Graves’ disease. None of the patients underwent completion thyroidectomy after initial lobectomy and/or received radioactive iodine (RI) therapy following the diagnosis of FTC. The mean total number of tissue blocks used for pathological diagnoses was 9.1 ± 4.1 (range, 3–20). Of these, the mean number of JGRDTC method blocks was 6.0 ± 2.8 (range, 1–12), and the mean number of additional blocks was 3.1 ± 2.0 (range, 1–8) (Table 2).

### 3.2. Pathological Diagnosis

Of the 86 patients, 7 cases of FTC were pathologically diagnosed based on the 3rd edition of the WHO classification criteria available at the time of surgery. The revised diagnoses based on the assessment of both JGRDTC method and additional blocks differed from the tentative diagnoses in just 4 patients based on the 3rd edition of the WHO classification, whose diagnoses were changed from follicular adenomas to FTC; therefore, a total of 11 cases of FTC were pathologically diagnosed. Thus, we could validate a statement reported by Lang et al. and Oh et al., who said that increasing the number of sampling increased the detection rate of capsular invasion in FTC [5,6].

Next, these diagnoses were revised using the 4th edition of the WHO classification criteria [3]. One patient was reclassified from follicular adenoma to FTC, and another patient from follicular adenoma to follicular tumor of uncertain malignant potential (FT-UMP), resulting in a total of six FTCs. The diagnoses of those with adenomatous goiter or PTC based on the assessment of JGRDTC method blocks were unchanged after additional blocks were analyzed (Table 3). Among the patients who were diagnosed with FTC (*n* = 6), three were classified as having the minimally invasive type and the other three were classified as having the encapsulated angioinvasive type, and no cases of the widely invasive type were reported. Of the 14 FTCs not included in this study, 12 were diagnosed as minimally invasive type and two as widely invasive type, according to the 3rd edition WHO Classification. A sub-classification of PTC variants showed one patient with the common (classic) type, one patient with the follicular type, and one patient with the macrofollicular type. These PTCs were incidentally found and were not detected with preoperative cytology.

The number of thyroidectomies performed during the study period (March 2016 to March 2017) was 837. Including the 11 patients with FTC as the revised diagnosis in the 86 patients in this study, there were a total of 25 patients with FTC during this period, which is 5.4% of the 466 patients with malignant thyroid tumors. Five cases were excluded from FTC by the review based on the 4th edition WHO criteria, resulting in a total of 20 FTCs, decreasing the incidence to 4.3% of the 461 cases of malignancy (Table 1).

### 3.3. Number of Blocks Used for Each Pathological Diagnosis

Table 4 shows the sizes of the main tumors in the revised diagnosis, following reclassification of 83 patients, excluding three cases of PTC, according to the 4th edition of the WHO classification criteria. A comparison between the FTC group, the FT-UMP/well-differentiated tumor of uncertain malignant potential (WDT-UMP) group, the follicular adenoma group, and the adenomatous goiter group revealed no statistically significant differences in tumor size, the number of JGRDTC method blocks, the number of additional blocks, or the total number of tissue blocks. Table 5 shows the numbers of JGRDTC method blocks and additional blocks assessed, and the total number of blocks per 1 cm of tumor based on the revised diagnoses of FTC, FT-UMP/WDT-UMP, follicular adenoma, and adenomatous goiter. Although the total number of blocks per 1 cm of tumor was slightly greater in the FT-UMP/WDT-UMP group, no significant differences were observed between the groups.

### 3.4. FTC and FT-UMP/WDT-UMP

Table 6 summarizes the details of FTC and FT-UMP/WDT-UMP cases based on the revised diagnosis of the primary nodule, following reclassification according to the 4th edition of the WHO classification criteria. Of the five patients who were diagnosed with FTC based on the JGRDTC method tissue blocks, two exhibited capsular invasion alone (cases 4 and 6), two exhibited capsular invasion and vascular invasion (<3 foci) (cases 2 and 5), and one exhibited vascular invasion alone (≥4 foci) (case 1). Case 3 was tentatively diagnosed with follicular adenoma; however, the diagnosis was changed to FTC upon revision. In this case, capsular invasion was observed only in the additional block. In case 8, an additional block contained findings suggestive of capsular invasion, resulting in the revised diagnosis being amended to FT-UMP. Six patients were tentatively diagnosed with FT-UMP/WDT-UMP; no unequivocal invasion was observed in the additional blocks in these cases, and none of these diagnoses were changed to FTC (cases 7 and 9–13). A microscopic thick tumor capsule was evident in two patients, one of whom had been diagnosed with FTC (case 2), and the other with FT-UMP (case 9). Thick fibrous capsules on gross examination, which can be indicative of FTC, were not observed in these cases. In all cases, the ultrasonographic findings were classified as Thyroid Imaging Reporting and Data Systems (TIRADS) 4A: undetermined. A nodule-in-nodule appearance, which can be indicative of FTC, was observed during ultrasonography in three cases (cases 4, 5, and 6), all of which were diagnosed with FTC based on the JGRDTC method blocks. Case 2 was complicated by the presence of an adenomatous goiter on the contralateral lobe, and case 7 was complicated by a micropapillary carcinoma on the contralateral lobe. Total thyroidectomy was performed in both cases. No patients underwent completion thyroidectomy and/or RI therapy following the diagnosis of FTC. At present, a mean period of 51 months (45–56 months) has passed for the examined 86 patients; all 86 patients were alive without recurrence of disease at the latest follow-up.

## 4. Discussion

When the diagnoses of 86 patients were reclassified according to the 4th edition of the WHO classification [3], only one follicular adenoma diagnosis (1/86: 1.2%) was amended to FTC following the analysis of additional blocks. A similar study by Hamza et al. previously examined 80 patients who had been diagnosed with follicular adenoma. Based on the pathological diagnoses of specimens from the entire capsule, they observed capsular invasion in three patients (3.8%). The diagnoses of these three patients were subsequently changed to minimally invasive FTC, whereas those of the remaining 77 patients remained unchanged, despite the additional examination cost of $4143 [10]. Oh et al. previously prepared additional blocks by vertically slicing the upper and lower ends of the transverse section of thyroid nodule specimens; this method led to an approximately two-fold increase in the number of diagnoses of capsular invasion; however, no change in the frequency of vascular invasion was observed [6]. Thus, it can be assumed that increasing the number of tissue blocks used for diagnosis could result in the increased frequency of diagnosis of suspected cases of capsular invasion. This has resulted in a more rigorous revised definition of capsular invasion. Various guidelines from organizations, such as the WHO, the American Joint Committee on Cancer (AJCC), the International Collaboration on Cancer Reporting (ICCR), and the College of American Pathologists (CAP), define capsular invasion as “complete penetration of tumor capsule” [11]. To prevent overdiagnosis and overtreatment of low-risk thyroid carcinoma, the concept of borderline lesions, such as non-invasive follicular thyroid neoplasm exhibiting papillary-like nuclear features (NIFTP), FT-UMP, and WDT-UMP, have been proposed [12,13,14]. The concept of borderline lesions has also been adopted in the 4th edition of the WHO classification criteria [3]. NIFTP is characterized by an encapsulated follicular pattern tumor with no invasion and delicate nuclear features of RAS-like PTC. In contrast, both FT-UMP and WDT-UMP are well-differentiated and encapsulated follicular neoplasms with questionable capsular invasion. When nuclear features of PTC are absent, a diagnosis of FT-UMP is made; when they are present, the diagnosis of PTC is changed to WDT-UMP. A recent study reported that many pathologists assess a lesion as FT-UMP if a questionable capsular invasion is observed [15]. In this study, questionable capsular invasion was observed in 7 patients based on the analysis of both JGRDTC method and additional blocks; of these, 5 were diagnosed as having FT-UMP. For the other two patients, findings of questionable capsular invasion and nuclear features equivocal to those of PTC were observed, leading to a diagnosis of WDT-UMP. However, even though additional blocks were examined for these patients, no definitive invasion that would otherwise justify changing the diagnosis to a malignant neoplasm was observed; therefore, those patients were not diagnosed with FTC.

Theoretically, in pathological diagnoses, only a very small portions of collected specimens (4–6 μm of the full-thickness of a 3–4 mm tissue block) can be microscopically examined. As a nodule is typically cut transversely or sagittally, the capsule is sliced in the direction tangential to the cut surface at both ends of the nodule. Hence, peripheral sections are cut perpendicular to the tumor capsule for pathological diagnosis. Yamashina reported a method of vertically resecting the entire capsule by cutting the nodule into wedges 2–3 mm in width [16]. Meanwhile, Seethala et al. recommend that pathological blocks be created by grinding the capsule of the nodule when making a diagnosis of NIFTP [12]. The absence of capsular invasion is a key factor in the diagnosis of NIFTP; however, creating blocks using these resection methods is substantially more costly and labor-intensive. Moreover, questionable levels of invasion may be found on pathological assessments if the capsule of the nodule is cut in a tangential direction. At our institution, the capsule is not cut in a tangential direction when resecting sections on both ends of the nodule when preparing tissue blocks. As shown in Table 4, at our institution, 5–8 blocks are normally prepared for the pathological diagnosis of a nodule with a size of approximately 4 cm. Additionally, as shown in Table 5, the total number of blocks per 1 cm of tumor for patients with FT-UMP and WDT-UMP were relatively, but not significantly, greater than the number of blocks per 1 cm of tumor for patients with FTC and follicular adenoma. As invasion was suspected in such cases, it can be interpreted that definitive findings could not be obtained, despite detailed examination of the additional tissue sampling.

Gross examination of the entire capsule is key to the effective preparation of tissue blocks. In general, FTC capsules are thicker and more irregular than those of follicular adenomas [17,18]. In all cases of FTC and FT-UMP/WDT-UMP, the ultrasonographic findings were classified in the TIRADS 4A: undetermined. In this category, ultrasonography patterns included “Simple neoplastic pattern,” “de Quervain pattern,” and “Suspicious neoplastic pattern” with a malignant frequency of 5–10% [19]. The TIRADS classification for diagnostic ultrasonography findings is generally suitable for the diagnosis of PTC but has a low diagnostic performance for FTC [20]. On the other hand, it has been found that a nodule-in-nodule appearance on ultrasonography or gross examination is also important in diagnosing FTC [21]. In the three cases in which the nodule-in-nodule feature was observed (cases 4, 5, and 6), capsular and vascular invasion had been observed in the JGRDTC method blocks; these cases were diagnosed as FTC. These findings suggest that it is important to prepare tissue blocks based on findings of suspected FTC on gross examination.

The prognosis of FTC varies greatly depending on the type of invasion [3,8,22]. Studies in Japan have found that the prognosis of minimally invasive FTC is favorable; for example, Sugino et al. reported that the 10-year survival rate was 98%, whereas Ito et al. reported that the mortality rate was as low as 1% during the observation period, which had a mean duration of 117 months [23,24]. In other countries, studies have also reported favorable prognoses of minimally invasive FTC with capsular invasion alone [25,26,27,28]. In addition, patients younger than 45 years of age, with no vascular invasion, reportedly have a favorable prognosis [29,30]. Oh et al. reported that among 342 patients with FTC, merely three patients (0.9%) exhibited synchronous distant metastasis and capsular invasion alone [6]. Stenson et al. reported that the risk of death from minimally invasive FTC was associated with combined capsular and vascular invasion, an age at surgery of ≥50 years, and male sex [31]. Lang et al. found that among 162 patients diagnosed with minimally invasive FTC, 7 (4.3%) developed distant metastases during the follow-up period. However, they pointed out that the mean number of tissue blocks per cm of tumor nodules was significantly lower for those 7 patients as compared to the others. They also discussed that the use of additional blocks at the time of specimen review might have revealed more extensive capsular or vascular invasion in those 7 patients. Moreover, while no distant metastasis was observed in 82 patients for whom 4 or more blocks/cm of tumor were examined, distant metastasis was observed in 7 out of 80 patients for whom three or fewer blocks/cm of tumor were examined [5]. In other words, this finding suggests that the diagnosis of FTC based on a larger number of blocks tends to indicate a favorable prognosis. An examination of vascular invasion and prognosis revealed that patients with focal vascular invasion without distant metastasis at presentation experienced no recurrence during the observation period (mean duration of 5.8 years) [28]. As explained above, it is possible to increase the number of diagnoses of FTC by detecting minimal invasion in an increased number of blocks; however, such an approach can result in overdiagnosis and overtreatment, which are unlikely to contribute to improving the prognosis of patients.

This study had several limitations. Firstly, since the mean observation period was short, being merely 51 months, the long-term prognosis remains unknown. Hence, certain patients who were diagnosed with follicular adenoma or adenomatous goiter may experience metastasis and may be diagnosed with FTC at a later date [32,33]. Secondly, none of the cases included in this study were diagnosed with widely invasive FTC, which can be associated with a poor prognosis. However, in cases of widely invasive FTC, tumor invasion outside the thyroid can frequently be confirmed via diagnostic imaging, such as through ultrasonography and computed tomography, and via gross examination of the cut surface of a resected specimen. Thus, an accurate pathological diagnosis of widely invasive FTC is straightforward when enough tissue blocks are prepared for evaluation. Thirdly, in this study, which included 86 patients, specimens of the entire capsule were not examined; therefore, the number of diagnoses of FTC may increase if an additional number of tissue blocks are examined.

In total, 20 patients were diagnosed with FTC between March 2016 and March 2017, following reclassification when cases with questionable invasion were excluded based on the 4th edition of the WHO classification criteria. This accounted for 4.3% of all cases of malignant thyroid tumors. This finding might suggest that revisions in the diagnostic threshold of invasion in the 4th edition of the WHO classification reduce numbers of FTC diagnoses despite meticulous sampling of tumor capsule which detects more capsular invasions that do not carry any impacts on the patient’s outcome.

The authors believe that this study has provided important insights into the optimal methods of preparing tissue blocks in thyroid practices with limited resources. In particular, several studies in Western countries have reported that in order to minimize the possibility of overlooking the presence of malignant tissue, it is ideal to search the entire capsule along with its surrounding tissue. However, examining the entire circumference of the capsule is hard in a practical scenario, even if the blocks are prepared using all resected segments. It is currently questionable whether increasing the frequency of minimally invasive FTC diagnoses based on the examination of a larger number of blocks can improve patients’ outcome [34,35]. In particular, in Japan, completion thyroidectomy and RI therapy are seldom performed for patients with minimally invasive FTC, as the clinical guidelines provided by the Japanese Association of Endocrine Surgeons do not recommend it in cases without distant metastasis; nevertheless, the prognosis of such patients is favorable [36,37]. There would be no difference in treatment methods even if minimally invasive FTC cases were misclassified as either follicular adenoma or FT-UMP, as all of these are curable with lobectomy alone. Since the opening of our hospital in 2006, we have not performed entire capsule sampling with encapsulated follicular tumors during pathological examination because it is not recommended in the Japanese clinical guidelines. Furthermore, among more than 5000 thyroid surgeries in our practice, there were no unexpected distant metastasis in benign diagnoses using the limiting sampling recommended by the JGRDTC. Based on these experiences in our practice, we concluded that our institution’s tissue sampling method (JGRDTC method) is appropriate because pathological diagnosis using entire capsule sampling is economically and technically challenging for hospitals in a community setting. Identification of high-grade carcinoma reflecting high risk of recurrence/metastasis and cancer death is essential in clinical practice, while entire sampling often identifies indolent tumors called minimally invasive FTC and FT-UMP/WDT-UMP, which creates overdiagnosis and overtreatment of thyroid carcinomas. When the 86 included patients were reclassified according the 4th edition of the WHO classification criteria, the number of those with FTC was found to be 6 instead of 11, and 7 patients met the criteria for biologically benign borderline lesions. This change to a benign diagnosis can be expected to be highly effective in alleviating patient anxiety about having thyroid cancer. Moreover, great care is required when preparing tissue blocks, to minimize the risk of overdiagnosis and overtreatment, and there is an unmet need to develop standards that can help make more accurate and efficient diagnoses.

## 5. Conclusions

In this study, an increasing the number of tissue blocks used for pathological diagnosis led to a limited (5.0%, 1/20) increase in the number of diagnoses of minimally invasive FTC based on the 4th edition of the WHO classification criteria. For examining a nodule with suspected FTC, the blocks were prepared from the maximum cut surface, avoiding tangential cuts. Sampling of 5–8 blocks from a 4 cm tumor is sufficient for accomplishing good thyroid pathology practice for accurate diagnoses of clinically significant FTCs. Ultrasonography and gross examination are key for ensuring the efficient preparation of tissue blocks.

## Figures and Tables

**Figure 1 cancers-13-01270-f001:**
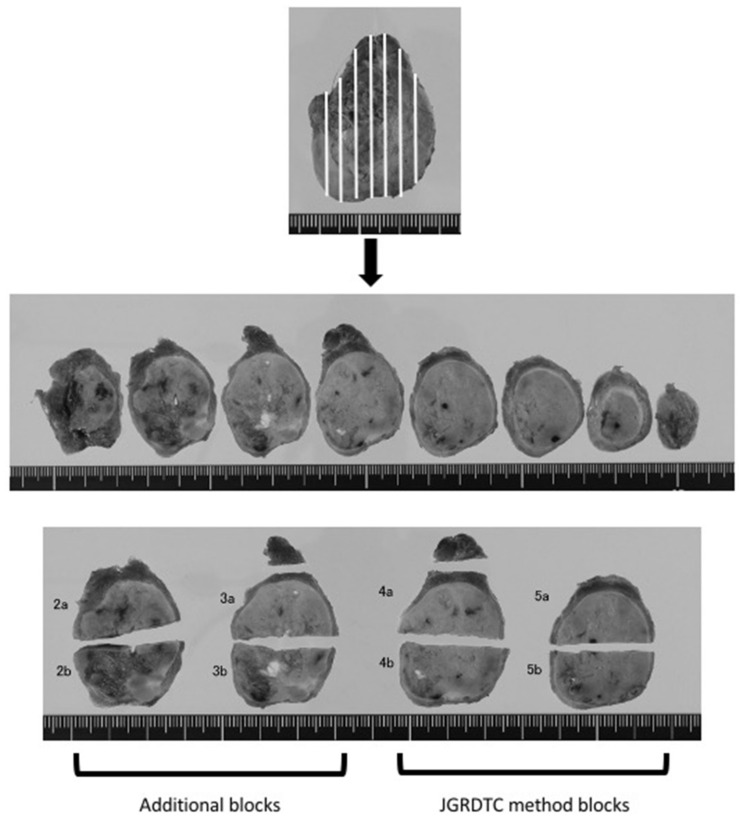
Examination of thyroid nodules based on recommendations of the Japanese Society of Thyroid Surgery. Multiple sagittal cuts at 3 mm intervals of a thyroid nodule, 37 mm diameter, after formalin fixation are shown (middle). In this study, four “JGRDTC method blocks” were prepared, comprising slices resected from the maximum cut surface (4a and 4b) and resected slices selected by gross examination (5a and 5b). In addition, 4 additional blocks were prepared (2a, 2b, 3a, and 3b) using 2 resected sections selected by gross examination (lower).

**Table 1 cancers-13-01270-t001:** Numbers of thyroidectomies performed and the frequencies of pathological diagnoses in the periods indicated.

Period	Number of Thyroidectomies Performed	Pathological Criteria	Malignant Thyroid Tumor Classification			Non-Malignant Tumor	Graves’ Disease and Others
			PTC	FTC	Other		
January 2015– December 2015	839	3rd WHO Classification	**469**	22	10	235	103
March 2016– March 2017	837	3rd WHO Classification	425	25 *	16	246 *	125
		4th WHO Classification	425	20 **	16	251 **	125

* Diagnoses based on the 3rd edition WHO classification using all available tissue sections. ** Revised diagnoses according to the 4th edition WHO classification criteria for only 86 cases in this study. Abbreviations: PTC, papillary thyroid carcinoma; FTC, follicular thyroid carcinoma; WHO, World Health Organization. Other malignant thyroid tumors include poorly differentiated carcinoma, anaplastic carcinoma, medullary carcinoma, and malignant lymphoma. Non-malignant tumors include benign and borderline thyroid tumor and hyperplastic nodules.

**Table 2 cancers-13-01270-t002:** Clinical background of the 86 patients and the numbers of tissue blocks assessed.

Background Factors	No (%)	Mean ± SD (Range)
Age at surgery		49.0 ± 17.8 (11–81)
≥55 years	37 (43.0)	
<55 years	49 (57.0)	
Sex		
Female	73 (84.9)	
Male	13 (15.1)	
Tumor size		4.1 ± 1.3 (1.6–8.9)
<2.0 cm	2 (2.3)	
2.0–4.0 cm	46 (53.5)	
>4.0 cm	38 (44.2)	
Operation		
Lobectomy/isthmectomy	72 (83.7)	
Total thyroidectomy	14 (16.3)	
RI therapy	0	
Number of tissue blocks		
JGRDTC method		6.0 ± 2.8 (1–12)
Additional sampling		3.1 ± 2.0 (1–8)
Total		9.1 ± 4.1 (3–20)

Abbreviation: SD, standard deviation; JGRDTC, Japanese General Rules for the Description of Thyroid Cancer; RI therapy, radioactive iodine therapy.

**Table 3 cancers-13-01270-t003:** Diagnoses of the 86 patients according to the 3rd and 4th edition of WHO classifications with JGRDTC method and revised pathological diagnoses.

Pathological Diagnosis	3rd Edition WHO Classification		4th Edition WHO Classification	
	JGRDTC Method	Revised Diagnoses	JGRDTC Method	Revised Diagnoses
**FTC**	7	11	5	6
FT-UMP/ WDT-UMP	-	-	6	7
FA	38	34	36	34
AG	38	38	36	36
PTC	3	3	3	3

Abbreviations: WHO, World Health Organization; JGRDTC, Japanese General Rules for the Description of Thyroid Cancer; FTC, follicular thyroid carcinoma; FT-UMP, follicular tumor of uncertain malignant potential; WDT-UMP, well-differentiated tumor of uncertain malignant potential; FA, follicular adenoma; AG, adenomatous goiter; PTC, papillary thyroid carcinoma.

**Table 4 cancers-13-01270-t004:** Tumor size by type and the number of tissue blocks assessed in each revised diagnosis.

Revised Diagnosis	FTC (*n* = 6)	FT-UMP/WDT-UMP (*n* = 7)	FA (*n* = 34)	AG (*n* = 36)	*p*-Value
Tumor size (cm)	4.1 ± 1.3 (1.9–6.3)	4.4 ± 1.1 (3.5–6.0)	4.3 ± 1.3 (2.3–8.9)	4.0 ± 1.3 (2.2–6.5)	0.701
Number of blocks by JGRDTC method	6.0 ± 3.2 (2–11)	8.1 ± 2.8 (4–12)	5.7 ± 2.7 (1–12)	5.8 ± 2.7 (2–12)	0.202
Number of additional blocks	2.8 ± 1.8 (1–7)	4.0 ± 1.8 (2–8)	2.9 ± 2.0 (1–8)	3.1 ± 2.0 (1–8)	0.606
Total number of blocks	8.8 ± 4.4 (3–15)	12.1 ± 4.1 (8–16)	8.6 ± 4.1 (3–18)	9.0 ± 4.4 (3–20)	0.226

The values are means, standard deviations, and ranges. The *p*-value was derived from one-way analysis and post hoc analysis. The *p* values are not significant. Abbreviations: FTC, follicular thyroid carcinoma; FT-UMP, follicular tumor of uncertain malignant potential; WDT-UMP, well-differentiated tumor of uncertain malignant potential; FA, follicular adenoma; AG, adenomatous goiter; JGRDTC, Japanese General Rules for the Description of Thyroid Cancer.

**Table 5 cancers-13-01270-t005:** Number of tissues blocks per 1 cm of tumor based on revised diagnoses.

Revised Diagnosis	FTC (*n* = 6)	FT-UMP/ WDT-UMP (*n* = 7)	FA (*n* = 34)	AG (*n* = 36)	*p*-Value
JGRDTC methodblocks/cm	1.4 ± 0.5	1.8 ± 0.5	1.3 ± 0.5	1.5 ± 0.5	0.071
Additional blocks/cm	0.6 ± 0.3	0.9 ± 0.4	0.7 ± 0.4	0.8 ± 0.4	0.268
Total blocks/cm	2.0 ± 0.6	2.8 ± 0.8	2.0 ± 0.7	2.3 ± 0.7	0.057

The values are means and standard deviations. The *p*-value was derived from a one-way analysis and post hoc analysis. The *p* values are not significant. Abbreviations: FTC, follicular thyroid carcinoma; FT-UMP, follicular tumor of uncertain malignant potential; WDT-UMP, well-differentiated tumor of uncertain malignant potential; FA, follicular adenoma; AG, adenomatous goiter; JGRDTC, Japanese General Rules for the Description of Thyroid Cancer.

**Table 6 cancers-13-01270-t006:** Clinicopathological characteristics of 13 patients with FTC and FT-UMP/WDT-UMP.

Case No	Age/ Sex	Tumor Size (cm)	Op	JGRDTC Method		Tentative Diagnosis	Additional		Revised Diagnosis	Total Blocks	Histological Appearance	TIRADSUS Findings
				Blocks	Invasion		Blocks	Invasion				
1	18/F	4.0	LO	4	vi	FTC-ea	2	vi	FTC-ea	6	thin capsule	TIRADS 4A
2	68/F	1.9	TT ^1^	3	ci+vi	FTC-ea	1	?	FTC-ea	4	thick capsule	TIRADS 4A
3	67/F	2.7	LO	2	no	FA	1	ci	FTC-mi	3	thin capsule	TIRADS 4A
4	66/M	4.3	LO	8	ci	FTC-mi	2	ci	FTC-mi	10	thin capsule	TIRADS 4A nod in nod
5	34/F	5.5	LO	8	ci+vi	FTC-ea	7	ci	FTC-ea	15	thin capsule	TIRADS 4A nod in nod
6	35/F	6.3	LO	11	ci	FTC-mi	4	ci	FTC-mi	15	thin capsule	TIRADS 4A nod in nod
7	29/F	3.5	TT ^2^	7	?	FT-UMP	2	?	FT-UMP	9	thin capsule	TIRADS 4A
8	34/F	3.7	LO	4	no	FA	4	?	FT-UMP	8	thin capsule	TIRADS 4A
9	14/F	4.0	LO	8	?	FT-UMP	8	?	FT-UMP	16	thick capsule	TIRADS 4A
10	68/M	4.4	LO	6	?	FT-UMP	4	?	FT-UMP	10	thin capsule	TIRADS 4A
11	62/M	4.4	LO	8	?	WDT-UMP	4	?	WDT-UMP	12	thin capsule	TIRADS 4A
12	27/F	4.8	LO	12	?	WDT-UMP	2	?	WDT-UMP	14	thin capsule	TIRADS 4A
13	76/F	6.0	LO	12	?	FT-UMP	4	no	FT-UMP	16	thin capsule	TIRADS 4A

Abbreviations: JGRDTC, Japanese General Rules for the Description of Thyroid Cancer; FTC, follicular thyroid carcinoma; FT-UMP, follicular tumor of uncertain malignant potential; WDT-UMP, well differentiated tumor of uncertain malignant potential; Op, operation (type); LO, lobectomy; TT, total thyroidectomy (due to complications of ^1^ contralateral adenomatous goiter and ^2^ bilateral papillary microcarcinoma); ci, capsular invasion; vi, vascular invasion; ?, questionable capsular invasion; no, no findings of invasion; FTC-ea, encapsulated angioinvasive follicular thyroid carcinoma; FTC-mi, minimally invasive follicular thyroid carcinoma; FA, follicular adenoma; TIRADS, Thyroid Imaging Reporting and Data Systems; US, ultrasound; nod in nod, nodule in nodule.

## Data Availability

The data presented in this study are available on request from the corresponding author due to ethical restrictions.

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
