# Peer review of "Additional Tissue Sampling Trials Did Not Change Our Thyroid Practice"

_cancers, 2021, doi:10.3390/cancers13061270_

Round 1
Reviewer 1 Report
The study aimed to investigate whether increasing the number of tissue blocks examined would lead to an increase in the number of FTC diagnoses. The topic is interesting and the information useful at this moment. I have some comments.
- the authors declared that the study is prospective. Absolutely not! This is a retrospective analysis of a series. Please, revise.
- the discussion discussion should be extent with a short discussion on the low performance of TIRADS in FTC (see Cancer Cytopathol. 2020 Apr;128(4):250-259. doi: 10.1002/cncy.22235).
- english should be imprived.
Reviewer 2 Report
The article by Shindo et al. Points the attention to, probably, the most interesting dilemma of thyroid pathology.
Follicular thyroid cancer are difficult to dignosed for several reasons.
The first reason is the absence of concrete preoperative diagnostic tecniques which can drive the intere process, the second is the focality of capsular or vascular infiltrations in the minimally invasive forms.The last is proved by many articles wich described the interobserver variability in the assesment of capsular infiltration.
Moreover, it is not possible, with the present techniques, to study, in a complete way, the tumoral capsule to establish the presence of infiltration.
Anyway I found the article confused.
The materials and methods are not completely clear, the casistic is short, the results are poor, with a very low percentage of patients changing diagnosis and a very short follow-up. The conclusions are not supported by significant data. So, the title results speculative.
For example:
in the materials and methods section:
Patients sub-section lines 96-97: the authors analyzed a retrospective period (January-December 2015) with 839 thyroid surgey performed, but they described 501 surgery for malignant tumors and 235 for benign tumors, in the table 1, too.
The study was prospective and I did not understand the meaning of this description in this section.
Tissue preparation and pathological evaluations sub-section: the authors declared that 86 nodules from patients were examined but they lack to describe why and in what way they selected the 86 patients among all patients in the period March 2016-March 2017.
Tissue preparation and pathological evaluations sub-sectionlines 119-123: the authors described the methods for “additional blocks” at our institution……”. It is not clear,probably this method is not standardized because they made a numbers of blocks on the basis of gross examination, so different for every nodule. Please better explain.
Furthermore, the authors never analyzed the complete capsule.
It should be interesting and very important, to standardize the method (on the basis of the tumor size I think) to make it repeatable and correlate its results with the results of the analysis of entire capsule.
In the discussion section:
lines 201-206: in the period March 2016-March 2017 among the 86 pazients studied (in which the authors found 11 FTC and 6 FTC +5 FTUMP/WDT-UMP) 25 effective FTC diagnosis. Please better explain.
In the discussion section:
lines 385-387 the authors mentioned a short follow-up. The lack of a long follow-up is the most important limitation of this study. You can assert the validity of this approach only in a speculative way.
Reviewer 3 Report
The authors present a study which addresses a controversial subject in endocrine pathology: the best way to sample the capsule of a thyroid tumor to achieve a correct pathological diagnosis. The design of the study is interesting and the comparison between the diagnostic criteria for follicular carcinoma in the previous and the current WHO Classifications adds further value to the MS.
Some point need to be clarified before the MS is suitable for publication
1) Can the authors explain the high number of FT-UMP and WDT-UMP (4th edition of WHO Classification) in their limited series of follicular tumors? The majority of those diagnoses seem to derive from benign lesions rather than FTC so probably the submission of additional blocks has increased the uncertainty instead of decreasing it
2) Which are the most important criteria the authors have taken in consideration for diagnosing UMP instead of FTC?
3) Did the authors apply immunohistochemical or molecular techniques to distinguish malignant from benign tumors?
4) In the Conclusions section the authors should state the importance of submitting additional blocks for refining the diagnosis of FTC since the follow-up of these patients, even without aggressive techniques (e.g. RAI administration), may keep the tumor under control
Round 2
Reviewer 2 Report
The authors answered to the questions and reviewed the article in the correct way, and I think it could be published.
Anyway, I continue to have some troubles.
It is extremely difficult to understand. I think it requires an editing of style
Reviewer 3 Report
The MS is now acceptable for publication